# Relating simulation studies by provenance—Developing a family of Wnt signaling models

**Kai Budde**[1]*, **Jacob Smith**[2], **Pia Wilsdorf**[1], **Fiete Haack**[1], **Adelinde M. Uhrmacher**[1]

**1** Institute for Visual and Analytic Computing, University of Rostock, Rostock, Germany, **2** Faculty of Computer Science, University of New Brunswick, Fredericton, Canada

* kai.budde@uni-rostock.de

**Data Availability Statement:** All relevant data are within the manuscript and its Supporting information files. Additional material is available at

## Abstract

For many biological systems, a variety of simulation models exist. A new simulation model is rarely developed from scratch, but rather revises and extends an existing one. A key challenge, however, is to decide which model might be an appropriate starting point for a particular problem and why. To answer this question, we need to identify entities and activities that contributed to the development of a simulation model. Therefore, we exploit the provenance data model, PROV-DM, of the World Wide Web Consortium and, building on previous work, continue developing a PROV ontology for simulation studies. Based on a case study of 19 Wnt/β-catenin signaling models, we identify crucial entities and activities as well as useful metadata to both capture the provenance information from individual simulation studies and relate these forming a family of models. The approach is implemented in WebProv, a web application for inserting and querying provenance information. Our specialization of PROV-DM contains the entities Research Question, Assumption, Requirement, Qualitative Model, Simulation Model, Simulation Experiment, Simulation Data, and Wet-lab Data as well as activities referring to building, calibrating, validating, and analyzing a simulation model. We show that most Wnt simulation models are connected to other Wnt models by using (parts of) these models. However, the overlap, especially regarding the Wet-lab Data used for calibration or validation of the models is small. Making these aspects of developing a model explicit and queryable is an important step for assessing and reusing simulation models more effectively. Exposing this information helps to integrate a new simulation model within a family of existing ones and may lead to the development of more robust and valid simulation models. We hope that our approach becomes part of a standardization effort and that modelers adopt the benefits of provenance when considering or creating simulation models.

## Author summary

We revise a provenance ontology for simulation studies of cellular biochemical models. Provenance information is useful for understanding the creation of a simulation model because it not only contains information about the entities and activities that have led to a simulation model but also their relations, all of which can be visualized. It provides additional structure by explicitly recording research questions, assumptions, and requirements

https://github.com/SFB-ELAINE/SI_Provenance_
Wnt_Family.

**Funding:** This work was supported by the
Deutsche Forschungsgemeinschaft (DFG, German
Research Foundation) (Grant no. SFB 1270/1—
299150580: K.B., F.H.; Grant no. 320435134: P.
W.) and by the Deutscher Akademischer
Austauschdienst (German Academic Exchange
Service) through the Research Internships in
Science and Engineering (RISE) program (Grant
no. 57467143: J.S.). The funders had no role in
study design, data collection and analysis, decision
to publish, or preparation of the manuscript.

**Competing interests:** The authors have declared
that no competing interests exist.

and relating them along with data, qualitative models, simulation models, and simulation
experiments through a small set of predefined but extensible activities.

We have applied our concept to a family of 19 Wnt signaling models and implemented
a web-based tool (*WebProv*) to store the provenance information from these studies. The
resulting provenance graph visualizes the story line of simulation studies and demon-
strates the creation and calibration of simulation models, the successive attempts of vali-
dation and extension, and shows, beyond an individual simulation study, how the Wnt
models are related. Thereby, the steps and sources that contributed to a simulation model
are made explicit.

Our approach complements other approaches aimed at facilitating the reuse and
assessment of simulation products in systems biology such as model repositories as well as
annotation and documentation guidelines.

## Introduction

Mechanistic, biochemical models are implemented and questioned to deepen the understand-
ing of biological systems. These models are usually the results of simulation studies that
include phases of refinement and extension of simulation models together with the execution
of diverse *in silico* (simulation) experiments.

A plethora of work has emerged over the last two decades to support the execution and doc-
umentation of simulation studies (e.g., modeling and simulation life cycles [1], workflows [2],
conceptual models [3]). Depending on the application domain, different modeling approaches
have their own documentation guidelines [4–6]. In the case of systems biology, the "Minimum
Information Requested in the Annotation of Biochemical Models (MIRIAM)" [7] and the
"Minimum Information About a Simulation Experiment (MIASE)" [8] are two community
standards used for documenting simulation models and corresponding simulation experi-
ments. A recent perspective by Porubsky et al. (2020) [9] looks at all stages of a biochemical
simulation study and at tools supporting their reproducibility. When looking at an entire sim-
ulation study and at the generation process of the included simulation model, these guidelines
provide some indication about what information might be useful for documenting a complete
simulation study as well as for establishing relationships between different simulation models.

This is of particular interest when several simulation models for a system under consider-
ation exist, offering different perspectives on the system, answering different questions, or
reflecting the data and information available at the time of generation. Model repositories such
as BioModels [10, 11], JWS Online [12], or the Physiome Model Repository 2 (PMR2) [13]
provide different means to retrieve and use simulation models. For example, querying the Bio-
Models database for biochemical and cellular simulation models that contain proteins such as
Wnt, Janus kinase (Jak), or mitogen-activated protein kinase (MAPK), which are associated
with corresponding signaling pathways, returns 22 simulation models for Wnt, 12 simulation
models for Jak and 139 simulation models for MAPK (as of January 2021). This already shows
that MAPK is an intensively studied signaling pathway. However, there is no way to easily
compare these simulation models or examine their relationships to each other. Sometimes
these relationships are represented in a model relationship map, such as the one created by
Ajmera et al. (2013) [14] for diabetes models. Tools such as BiVeS [15] are helpful to compare
different versions of one particular simulation model, but comparing different models—even
of the same system—is a difficult task because the syntax of these models (e.g., the names of
the species), as well as their reactions, might be completely different. Instead of analyzing the

similarities and differences in the specifications of simulation models to infer possible relationships between these simulation models, we will focus on context information, such as how a simulation model has been generated.

Particularly, larger models are usually not built from scratch [16]. In general, simulation models are the outcome of extensive as well as interactive model and data generation activities. These include, in addition to executing various simulation experiments and successive model refinements, the adaptation of already existing models, for example, by composition or extension [16–18]. Therefore, the complexity of a model grows over time as researchers add parts to the model or refine it. Keeping track of these generation processes is the subject of provenance.

Provenance provides "information about entities, activities, and people involved in producing a piece of data or thing, which can be used to form assessments about its quality, reliability or trustworthiness" [19]. Thus, it can be applied to many fields of science, art, and technology, including biochemical and cellular simulation models. By exploiting standardized provenance data models, such as PROV-DM, this information is presented in a structured, queryable, and graphical form [20, 21]. In addition to providing crucial information about the generation of an individual simulation model and, thus, facilitating its reuse, provenance can be applied to reveal and exploit relationships between different simulation models.

In this publication, we will identify and structure relevant information needed to capture the provenance of simulation studies and elucidate how a family of simulation models can be established through relating the models to each other. As a case study, we will concentrate on 19 simulation studies of the Wnt/β-catenin signaling pathway. Among the different Wnt signaling pathways, the canonical Wnt or Wnt/β-catenin signaling pathway is the most intensively studied one, *in vitro* [22] as well as *in silico* [23]. This pathway is considered to be one of the key pathways in development and regeneration, including cell fate, cell proliferation, cell migration and adult homeostasis [24, 25]. In deregulated forms, it is involved in human cancers and developmental disorders [26, 27].

Our case study refers to the Wnt/β-catenin signaling pathway only, which we call Wnt throughout the text. We will present and use our web-based tool *WebProv* to store, display and query provenance information from these simulation studies. Different queries and analyses of the family of 19 Wnt simulation models will then be used for finding further insights into the family of Wnt signaling simulation models.

## Materials and methods

### Provenance of simulation models

**Provenance data model.**   We consider the types and relations defined by the PROV data model: PROV-DM [19]. PROV-DM includes the following types: `entity`, `activity`, `agent` as well as the following relations: `WasGeneratedBy`, `Used`, `WasInformedBy`, `WasDerivedFrom`, `WasAttributedTo`, `WasAssociatedWith`, `ActedOnBehalfOf`. Provenance information is usually depicted as a directed, acyclic graph where the arrowheads show towards the sources or predecessors of an entity, activity, or agent—thus, depicting its origin. For our case study, we are only focusing on the types `entity` and `activity` as well as on the relations `WasGeneratedBy` and `Used`. The reasons for our decision will be explained in the Results and discussion section.

**Provenance ontology for simulation studies.**   Recently, Ruscheinski et al. (2018) [20] have applied PROV-DM for capturing provenance information from entire simulation studies and initiated a definition of a PROV-DM ontology for these studies. Important ingredients of

this ontology have been identified to be "a) specific types of entities (e.g., data, theories, simulation experiments, and simulation models), b) specific roles between these entities (e.g., used as input, used for calibration, used for validation, used for adaptation, used for extension, used for composition), c) specific refinement of activities (i.e., successive refinement of activities down to a level where simulation experiment specifications define activities and thus are ready to be executed), and d) specific inference strategies (e.g., warnings if the same data has been used both by calibration and validation activities, or the option to reuse validation experiments among model descendants to check consistency)" [20]. The adaptation and application of this ontology for capturing the essential information of our case study is presented in the Results and discussion section.

**Collecting provenance information.** In order to gather all relevant information, the publications as well as the supporting materials—as they often contain model and experiment descriptions—were read thoroughly. Referenced publications were checked, as well, whenever they appeared to be important for the development of the simulation model. All information that resembled provenance entities were marked. While reading a study, a first sketch of a possible provenance graph was made. Afterwards, a revision of all markings helped to finalize the graph and to remove duplicate entities. Often, authors described their simulation study chronologically, which made it easy to determine the path of its development, but sometimes, the connections of the entities had to be inferred from the context. In general, tracing provenance information from an entire simulation study in retrospective involved some interpretation of the results presented in the publication.

## Implementing the PROV-DM ontology: WebProv

We have developed *WebProv*, a web-based tool that can be used to store, access, and display provenance information from simulation studies. It allows one to insert and query provenance information based on a web interface as frontend and a graph-based database as backend. The frontend uses Vue, a popular JavaScript reactivity system, along with D3.js, a JavaScript visualization library, to create the front-end visualizations and power the node/relationship editor. As scalability was not a concern when designing the tool, all graph data is sent to the frontend when the website is first opened, allowing the frontend to perform approximate string matching and explore the entire graph without additional queries to the database. Although this reduces the responsibilities of the back-end system, the backend still provides an interface for loading different types of nodes, updating data and importing/exporting JSON data from Neo4j. Furthermore, the backend allows one to load in a set of nodes and relationships from JSON into Neo4j on startup to initialize the database.

The tool can also be installed locally for testing and replicability purposes. Details about its installation, as well as the code, can be found on GitHub. An informative video showing the usage of *WebProv* is also on YouTube.

**Provenance nodes.** The main concept of *WebProv* is the Neo4j Provenance Node and the dependency graph created from related Provenance Nodes using Neo4j relationships. A Provenance Node represents an entity or activity and, therefore, must have a classification (e.g., Simulation Model or Building Simulation Model) which defines the types of relationships that can be formed with other nodes depending on our PROV ontology. For example, the Simulation Model entity can only be created by a Building Simulation Model or a Calibrating Simulation Model activity (see the Results and discussion section for details). These classifications can be easily changed or extended if necessary. Additionally, `Study` nodes store information about a particular study (a reference to a study and the name of the signaling pathway it is concerned with) and group a set of Provenance Nodes together. Finally, `InformationField` nodes

allow us to attach zero or more key–value pairs to a Provenance Node to store further information. In our case, we describe which information should be entered depending on the entity type in the Results and discussion section.

**Queries.** *WebProv* allows two types of queries: text queries and queries in Cypher—Neo4j's query language. The text query field will perform a fuzzy search of the data contained within the nodes. If successful, a set of nodes are returned that contain the given phrase and the user can choose to add any of these nodes to the graph. Alternatively, the Cypher field passes the query as a string to the backend which forwards it to the Neo4j database. Since arbitrary queries can be performed, when the results are returned, the frontend attempts to parse the results using io-ts as a Provenance Node. If successful, these nodes are automatically shown on the graph. This method allows more complex queries, in particular, subgraphs to be extracted. Thus, structural information can be accessed.

## Wnt signaling models

A comprehensive list of published simulation studies that deal with or include the Wnt signaling pathway is found in Table 1. Out of the 31 simulation studies ([28–58]), which we have found, we have chosen to collect provenance information from 19 studies, shown in bold in Table 1. Some of these Wnt models have already been discussed in previous reviews [23, 59]. We have included all Wnt simulation studies where simulation models were stored in BioModels (6 studies) as well as all Wnt simulation studies published by our group (4 studies). The remaining nine studies were selected randomly.

All models include Wnt ligands and Wnt receptors (implicitly or explicitly) as well as the Wnt signal transducer protein β-catenin. There are two exceptions: the simulation model by Sick et al. 2006 [31] contains only Wnt and its antagonist Dkk and the model by Rodríguez-González et al. (2007) [33] lacks β-catenin. The number of species without considering compounds or different attributes or states, such as the phosphorylation state, ranges from 2 (in [31]) to 30 (in [53]). The dimension of a system may be higher if a model contains compounds or different states of the species. A schematic representation of the Wnt signaling model including relevant species and interactions from all 19 surveyed studies is shown in Fig 1.

## Results and discussion

In order to provide useful information about a set of simulation models as a kind of family, we need to answer the questions about which information regarding these models and their development processes are needed and how to describe them. Based on our earlier work on provenance of simulation models, we refine a specialization of the PROV Data Model (PROV-DM) and, thus, define a PROV ontology that is capable of both relating simulation models and reporting their generation processes. We also examine the level of detail, or granularity, that is necessary to capture relevant information of the provenance of simulation studies.

First, we will introduce and discuss our specialization of PROV-DM for cellular biochemical simulation models. This part contains descriptions and examples of all entity and activity types used in our provenance data model. Fig 2 and Table 2 provide overviews of these entity and activity types and should be consulted when skipping the first section.

Second, we will discuss our findings, applying our specialization of PROV-DM and demonstrate the relationships as well as specific features of the provenance information from the 19 Wnt simulation studies covered in this publication.

**Table 1. Wnt simulation models (as of Feb. 1, 2021) with those included in this study printed in bold.**

| Study | BioModels | MA | SA | Scale | Add. Compartm. | Add. Pathways/Models |
|-------|-----------|-----|-----|-------|----------------|----------------------|
| [28] | ✓ | ODE | det | SC | – | – |
| [29] | – | ODE | det | SC | – | – |
| [30] | – | ODE | det | SC | – | – |
| [31] | – | PDE | det | TOL | – | – |
| [32] | ✓ | ODE | det | SC | Nuc | MAPK/ERK |
| [33] | – | ODE | det | SC | – | Notch |
| [34] | ✓ | ODE | det | SC | – | – |
| [35] | – | ODE | det | SC | – | – |
| [36] | ✓ | ODE | det | SC | Nuc | Notch, MAPK/ERK |
| [37] | – | PDE | det&stoch | MC | – | E-cadherin |
| [38] | – | ODE | det | SC | – | – |
| [39] | – | ODE | det&stoch | MC | – | Cell cycle, E-cadherin |
| [40] | – | ODE | det | SC | – | – |
| [41] | – | ODE | det | SC | – | – |
| [42] | – | PDE | stoch | MC | – | – |
| [43] | – | ODE | det | SC | – | MAPK/ERK |
| [44] | – | PDE | det&stoch | MC | – | Notch |
| [45] | – | ODE | det | SC | – | – |
| [46] | ✓ | Rule | det&stoch | SC | Nuc | Cell cycle |
| [47] | – | ODE | det | SC | Nuc | – |
| [48] | – | ODE | det | SC | Nuc | Notch |
| [49] | – | ODE | stoch | SC | Nuc, GA | E-cadherin |
| [50] | – | ODE | det | SC | Nuc | – |
| [51] | – | Rule | stoch | SC | Nuc, Mem, LR | ROS |
| [52] | – | ODE | det | SC | Nuc | – |
| [53] | ✓ | ODE | det | SC | – | MAPK/ERK, PI3K/Akt |
| [54] | – | Bool | det | SC | – | PI3K/AKT, MAPK/ERK, Rho/Rac |
| [55] | – | ODE | det | SC | – | – |
| [56] | – | Rule | det | SC | Nuc, End, Mem, LR | – |
| [57] | – | Rule | stoch | SC | Nuc | ROS |
| [58] | – | ODE | det&stoch | MC | – | Cell cycle, Hippo |

A list of published simulation studies of the Wnt signaling pathway is presented showing the references, the availability of the simulation models in BioModels, the modeling approaches (MA), the simulation approaches (SA), the scale of the models, additional compartments as well as additional pathways or sub-models included. The authors of the studies printed in bold are: [28]: Lee et al. (2003), [29]: Krüger and Heinrich (2004), [30]: Cho et al. (2006), [31]: Sick et al. (2006), [32]: Kim et al. (2007), [33]: Rodríguez-González et al. (2007), [34]: van Leeuwen et al. (2007), [35]: Wawra et al. (2007), [36]: Goldbeter and Pourquié (2008), [39] van Leeuwen et al. (2009), [41]: Mirams et al. (2010), [45]: Kogan et al. (2012), [46]: Mazemondet et al. (2012), [48]: Wang et al. (2013), [49]: Chen et al. (2014), [51]: Haack et al. (2015), [53]: Padala et al. (2017), [56]: Haack et al. (2020), and [57]: Staehlke et al. (2020). The BioModels IDs of the simulation models available in BioModels are: [28]: BIOMD0000000658, [32]: BIOMD0000000149, [34]: MODEL2001090001, [36]: BIOMD0000000201, [46]: MODEL1303140000, [53]: BIOMD0000000648. The modeling approaches (MA) are: ODE-based (ODE), PDE-based (PDE), rule-based or reaction-based (Rule), and Boolean network model (Bool). The simulation approaches (SA) are: det (deterministic), stoch (stochastic). The scale may be single cell (SC), multi cell (MC) or at a more abstract tissue/organ level (TOL). Every simulation model contains at least one compartment—usually the cytosol. We also denote additional compartments where reactions may take place and where some species may shuttle into or out of: Nucleus (Nuc), Membrane (Mem), Endosome (End), Golgi apparatus (GA), Lipid Raft (LR). Models without shuttling are considered to have only one compartment even though they describe processes in different places, for example, in the cytosol, nucleus and at the cell membrane.

## Further steps towards a PROV-DM ontology for cellular biochemical simulation models

We have revised and refined the specialization of PROV-DM, which was introduced by Ruscheinski et al. (2018) [20]. For capturing provenance information from simulation

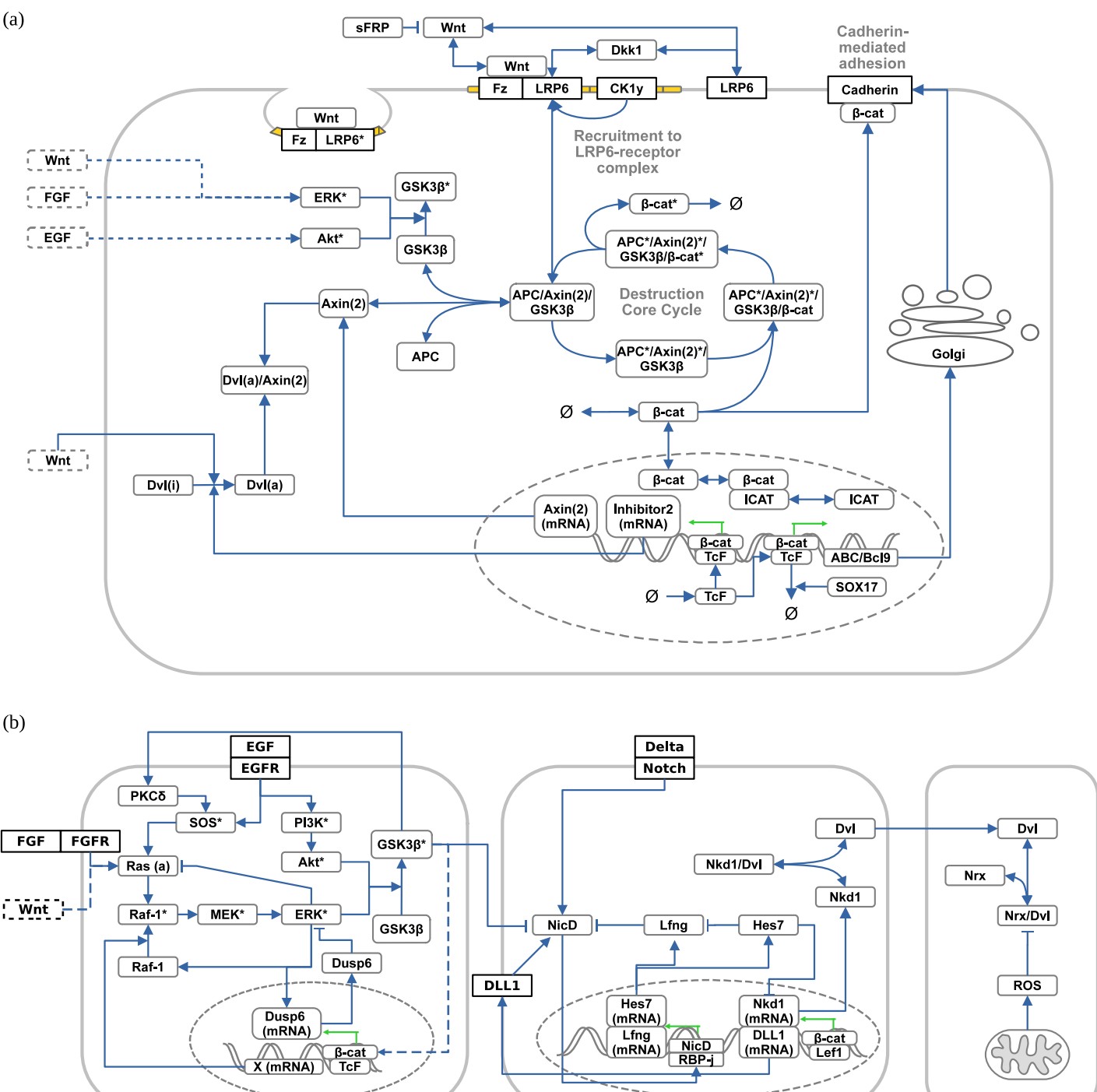

**Fig 1. Combined overview of all qualitative Wnt (sub-)models found within the 19 Wnt simulation studies.** Depicted are the components (species, compartments, and reactions) of the canonical Wnt signaling pathway (a) and its crosstalk with other signaling pathways (b) found within the 19 Wnt simulation studies. Note that the overview is a simplified and condensed representation. Interactions are simplified and some components of the submodels that do not directly affect the Wnt signaling pathway are omitted. Activated/phosphorylated proteins are indicated by (*). Inactive/unphosphorylated states of proteins have been omitted when possible. Submodels involving membrane-mediated processes, such as receptor/ligand interactions, destruction complex recruitment and endocytosis [31, 45, 51, 56], or cadherin-mediated cell adhesion [34, 39, 49] are incorporated in (a). Submodels involving crosstalk with ERK/FGF/PI3K/Akt [32, 36, 53], Notch [33, 36, 48], and ROS/Dvl-mediated pathways [51, 57] are shown in the lower panels of (b), respectively.

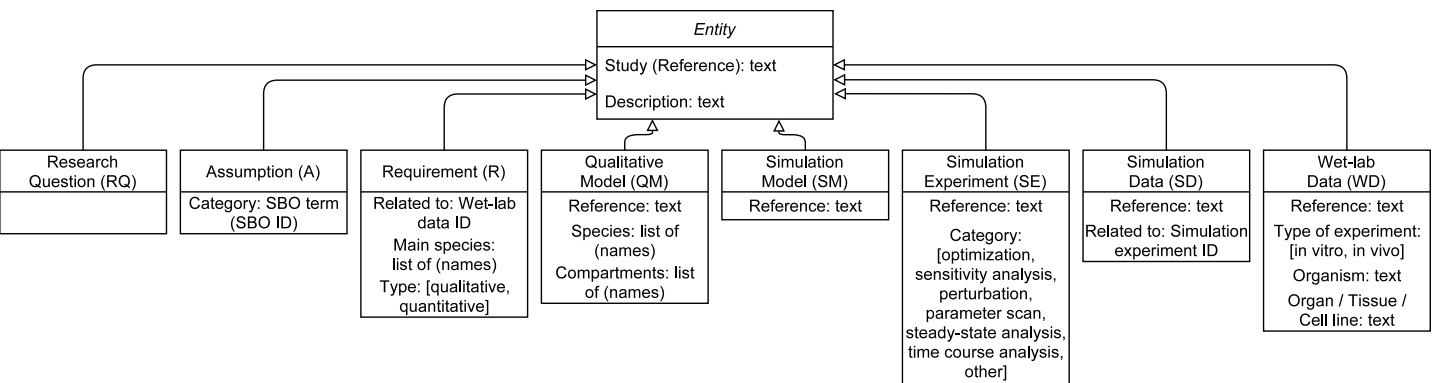

**Fig 2. UML class diagram of provenance entities in *WebProv*.** We have identified the following entities to be useful for providing provenance information from simulation studies in the field of systems biology: Research Question, Assumption, Requirement, Qualitative Model, Simulation Model, Simulation Experiment, Simulation Data, and Wet-lab Data. The requested information for each entity type is kept minimal for demonstration purposes and can easily be extended. The *Study (Reference)* contains information of the publication, for instance, "Lee et al. (2003)" and determines which study an entity belongs to. The *Description* contains a brief explanation of a particular entity and may be a cited text from the publication. Furthermore, entity references should ideally consist of a digital object identifier (DOI) to make the artifact associated with the particular entity unambiguously accessible. Additional information can always be entered in the "Further Information" part of *WebProv*.

**Table 2. Entities, activities and allowed relations in our PROV-DM specialization.**

| Entity | *wasGeneratedBy* (Activity) |
|---|---|
| Research Question (`RQ`) | – |
| Assumption (`A`) | – |
| Requirement (`R`) | – |
| Qualitative Model (`QM`) | – |
| Simulation Model (`SM`) | `BSM` \| `CSM` |
| Simulation Experiment (`SE`) | `CSM` \| `VSM` \| `ASM` |
| Simulation Data (`SD`) | `CSM` \| `VSM` \| `ASM` |
| Wet-lab Data (`WD`) | – |
| **Activity** | ***used* (Entity)** |
| Building Simulation Model (`BSM`) | `RQ` \| `SM`, {`RQ` \| `A` \| `R` \| `QM` \| `SM` \| `SE` \| `SD` \| `WD`} |
| Calibrating Simulation Model (`CSM`) | `SM`, {`A` \| `R` \| `SD` \| `WD`} |
| Validating Simulation Model (`VSM`) | `SM`, {`A` \| `R` \| `SD` \| `WD`} |
| Analyzing Simulation Model (`ASM`) | `SM`, {`A` \| `SD` \| `WD`} |

Left column: Specified PROV-DM entity and activity types used for capturing provenance information from simulation studies. Right column: Relations of PROV-DM used for capturing provenance information from simulation studies as well as allowed connections of entities/activities from the first column. The PROV relation *wasGeneratedBy* connects `entities` with `activities`; *used* connects `activities` with `entities`. The Research Question, Assumption, Requirement, Qualitative Model, and Wet-lab Data entities are included in the provenance graph without their origins, thus without an activity generating them. The generation of the Simulation Model, Simulation Experiment, and Simulation Data are explicitly shown in the provenance graphs. For example, a Simulation Model can be created or updated based on a Building Simulation Model or Calibrating Simulation Model activity—the alternative is denoted by "|". Regarding the activities, each activity has at least one entity it depends on (Research Question or Simulation Model). Other entity types are optional and several or none of each of them may be used by one particular activity—denoted by {...} in the extended Backus–Naur form (EBNF).

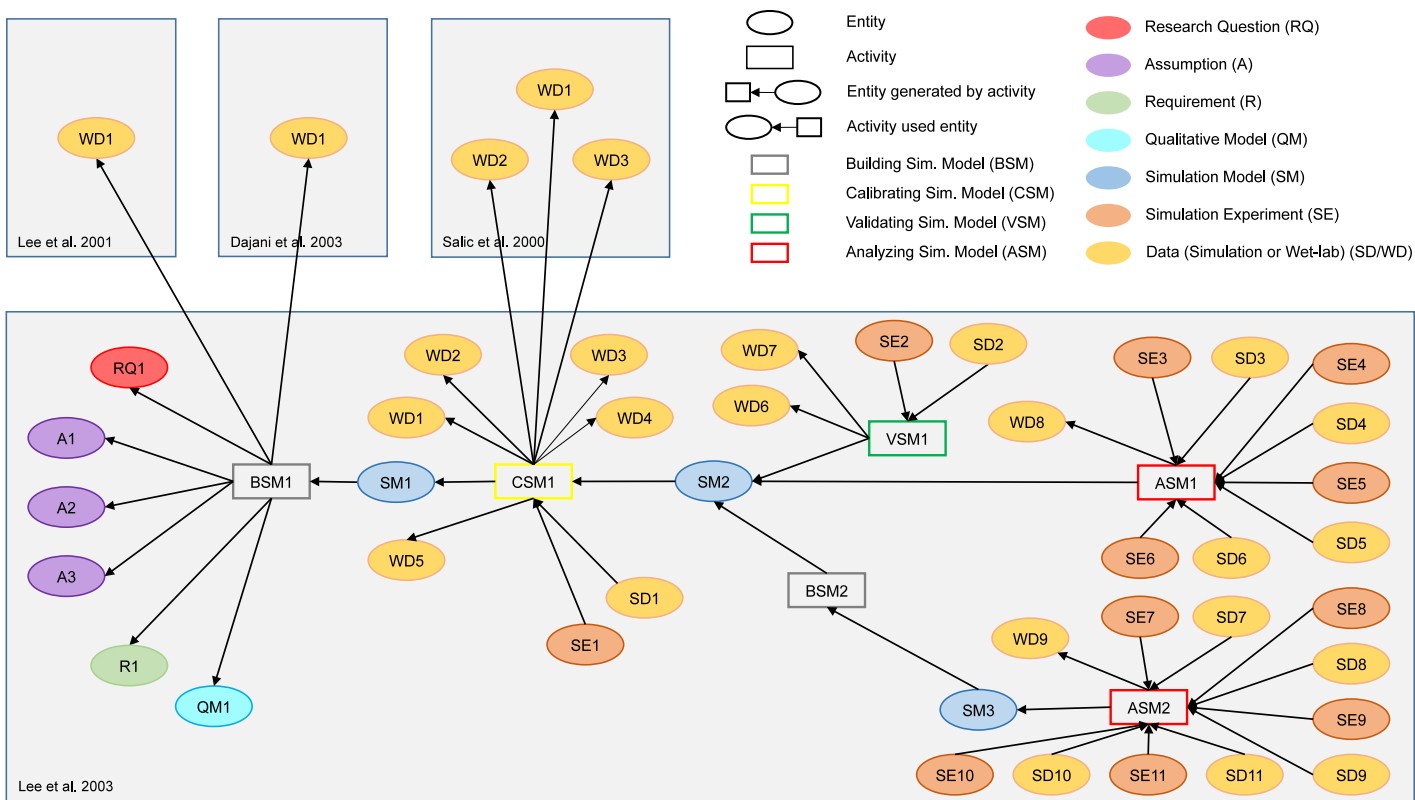

**Fig 3. Provenance graph of the study by Lee et al. (2003) [28].** Besides the entities and activities that make up the provenance information from the study (see legend), additional entities from three other studies [64–66], which were used by Lee et al. (2003), are shown. The colors of the ellipses show different entity types, the borders of the rectangles visualize different activity types. The gray areas separate the individual studies. The graph displays, for example, that the Building Simulation Model activity `BSM1` used, among others, the entity `WD1` of type Wet-lab Data from Lee et al. 2001 [64]. This activity then generated the Simulation Model `SM1`.

studies of cellular biochemical simulation models and relating these, we are defining and using a) specific types of entities and activities and b) specific relations with their roles and constraints. During the process of collecting provenance information from the studies, we identified the types and relations as well as information that was useful for describing them. Our final set of entities, activities, and relations is shown in Table 2. Each entity and activity has already been mentioned for provenance, modeling or documentation purposes, or experiment design of simulation studies [1, 3–5, 8, 20, 21, 60–63], but they have not all been used together.

In the following section, we will describe these entities, activities, and relations and discuss the information that should be included in *WebProv*. For each entity and activity, we will show examples of our specialization with provenance information obtained from the provenance graph of the simulation study by Lee et al. (2003) [28], shown in Fig 3, which also includes additional entities from three other studies [64–66].

**Provenance entities.** For every Provenance Node (entity or activity), we require the following details to be provided: a) (PROV-DM) Type, b) Study (Reference), c) Description.

The (PROV-DM) Type declares the type of entity or activity. The Study (Reference) contains the name of the study a Provenance Node belongs to. The Description contains some textual explanation of the Provenance Node. For some entities, we are asking for further information, as seen in Fig 2, which we will elaborate on.

**Research Question** [RQ]:

The research question (or research objective or problem formulation) determines the goal of the research presented in a publication. For simulation studies, it typically forms the starting point of the modeling and simulation life cycle [1, 67] and is key to interpreting its outputs such as simulation data or a simulation model.

As for the provenance example shown in Fig 3, Lee et al. (2003) questioned in RQ1 the necessity of "the two scaffold proteins, APC and Axin" (for Wnt signaling) and whether "their roles differ" [28]. This research question determines a minimum number of model constituents and guides the modeler towards possible simulation experiments to be executed.

**Assumption** [A]:

We define assumptions of a simulation model to be all statements that refer to abstractions or specializations of the described model. Assumptions typically deal with the input of a model (e.g., *assume the concentration of x to be constant* or *let the initial value of y be . . .*) and may set the boundaries of the system under consideration or partially explain the thoughts behind a simulation model—always with the research question in mind.

In order to facilitate the analysis of assumptions, we adopted the Systems Biology Ontology (SBO) [68] to categorize the assumptions. SBO provides "structured controlled vocabularies, comprised of commonly used modeling terms and concepts" [69] and is primarily used to "describe the entities used in computational modeling (in the domain of systems biology)" [68]. By using SBO, we are trying to answer which part of the model contains assumptions rather than what was assumed.

In the provenance example, three assumptions with three different categories could be identified. A1, for instance, reads "Dsh, TCF, and GSK3β are degraded very slowly, we assume that their concentrations remain constant throughout the timecourse of a Wnt signaling event" [28] and was matched to ID 362 (Concentration conservation law) of SBO.

**Requirement** [R]:

Requirements define properties that the results of a simulation model need to show. These may be used for the purpose of calibrating or validating a simulation model. They also direct the modeler towards adaptation of a model if the requirements are not met. We do not consider other kinds of requirements (e.g., the need of using specific tools or approaches in performing a simulation study).

Typically, simulation data needs to be compared with real-world data—in our case wet-lab data. These real-world measurements determine the species of interest which should be part of the Requirement entity. Therefore, we record the main species considered by the requirement as well as its type (either qualitative or quantitative) and connect the Requirement to the wet-lab data it relates to. The list of main species will make it easier to compare, interrelate and reuse simulation models as they determine the focus of the model.

We were able to identify one requirement R1 in the provenance example of Lee et al. (2003). The quantitative requirement that "Axin stimulates the phosphorylation of β-catenin by GSK3β at least 24,000-fold" [28] actually refers to the wet-lab data WD1 obtained in another study by Dajani et al. (2003) [65]. Its main species are Axin, β-catenin, and GSK3β.

**Qualitative Model** [QM]:

We define the qualitative model to be a network diagram, such as a reaction scheme (chemical reaction network diagram), which contains the entities of the system (e.g., species) and their interactions. This diagram may be presented in a formal (e.g., using the Systems Biology Graphical Notation (SBGN) [70] or a Boolean network diagram [71]) or informal way. All textual descriptions of a simulation model that do not include quantitative information (e.g., reaction rate constants, initial values) can also be part of the qualitative model. It should be noted

that the qualitative model is also called conceptual model [72] sometimes, whereas in other publications, the qualitative model forms part of the conceptual model [3].

We record a reference to the qualitative model, which, for example, could be a reference to a figure in the publication, or, ideally, a DOI. Furthermore, we denote a list of species and compartments used in the model. Multiple compartments require a shuttling of species from one compartment to another one and every compartment should, ideally, have an area (for the transfer rate or concentration in two-dimensional compartment) as well as a volume (for a three-dimensional compartment) [73].

In our provenance example of Lee et al. (2003), QM1 contains a qualitative model in the form of a chemical reaction network diagram which can be directly accessed via a DOI. It represents reactions of the species Wnt, Dsh, GSK3, Axin, APC, β-catenin, and TCF within a cell extract.

**Simulation Model** [SM]:

The simulation model is the actual mathematical or computational model [74] that can be executed by a suitable tool. In most cases of our domain, the simulation model contains equations (for ODE/PDE systems) or, in some cases, reaction rules (for rule-based systems). An integral part of these quantitative simulation models are the parameter values as well as the initial condition. The simulation model could also be described in another form (e.g., in a quantitative process algebra [75, 76] or with a combination of multiple formalisms [77]). Formal approaches to describe a system through qualitative models (e.g., Boolean models [78] or Petri nets [79]) come with their own means of analysis and are assigned to the Simulation Model entity as they are executable models. Usually, a parameter table complements the description of the simulation model and gives information about the parameter values and their origin.

A new Simulation Model entity is created whenever the reaction network changes or after a simulation model has been calibrated, which typically means that the set of parameter values and the initial condition have been (re-)defined. A validation activity (by itself) does not alter the simulation model, although a failed validation activity is likely to induce a change of the simulation model (see, for instance, [56]).

Again, we are relying on a reference of the simulation model for accessing it. It should be a link to the simulation model in Biomodels or a DOI to the description of the simulation model. Ideally, it is presented in a structured and widely accepted format such as SBML [80] or CellML [81].

As for the provenance example, the calibrated simulation model of Lee et al. (2003), SM2, can be found in BioModels.

**Simulation Experiment** [SE]:

The simulation experiment is an execution of the simulation model. Ideally, it can be linked to a complete experiment specification (e.g., as a SED-ML [82] or SESSL [83] file or simply as the execution code in a general purpose programming language) and to documentation in a standard format that applies reporting guidelines such as MIASE for simulation experiments [8]. Different simulation experiments might be used for the analysis, calibration, and validation of a simulation model.

To further structure the set of applied simulation experiments, we distinguish simulation experiments by whether they are used for optimization, sensitivity analysis, perturbation, parameter scan, steady-state analysis, or time course analysis. This list is neither complete nor are the categories disjoint, and, given a different set of simulation studies, they will likely be subject to renaming, extension, and refinement.

We have defined optimization experiments to be all experiments where an implicit or explicit objective function is used. This includes parameter estimation as well as manual parameter fitting experiments. If these succeed, the Calibrating Simulation Model activity will

produce a new (calibrated) Simulation Model. In a sensitivity analysis, more than one parameter value is changed and some kind of sensitivity coefficient is calculated. We have interpreted experiments where the value of one (or more) parameter is changed to another (just one) value, for example, to mimic a knock-out experiment as perturbations. Parameter scans include the variation of at least one parameter value within a given range. A steady-state analysis is aimed at identifying the steady state of a system. We refer to time course analysis to be the analysis of trajectories without varying parameter values.

Eventually, an ontology about the various experiment types and analysis methods and their use in simulation studies will be crucial as simulation experiments play a central role in the provenance of simulation models. This would also help to exploit the provenance information effectively, for example, for automatically generating simulation experiments [84].

In the case of Lee et al. (2003), different simulation experiments have been executed. For example, `SE1` contains a parameter scan in order to validate the simulation model. However, no further details are given in the paper, therefore, no reference could be included in the entity (the reference is "not available").

Note that we have not included a Wet-lab Experiment entity. Our focus is on the result of the Wet-lab Experiment (i.e., the Wet-lab Data) and its role within the simulation study (e.g., being used in a Building Simulation Model, Calibrating Simulation Model, or Validating Simulation Model activity).

**Simulation Data** [`SD`] **and Wet-lab Data** [`WD`]:

Data is the result of an experiment. In our case, it can either stem from wet-lab or simulation experiments. It includes a reference to a plot or table or, ideally, to a database containing the raw data. Because simulation data is generated by a simulation experiment, a link needs to be established. In the case of a simulation experiment that serves the role of validation, information about the success of a validation facilitates the interpretation of the simulation model and further activities based on the simulation model. As no independent Wet-lab Experiment entity is supported, details about the wet-lab experiment can be summarized in the description of the Wet-lab Data or by referencing, for example, a research protocol on Protocols.io [85]. The type of the wet-lab experiment (*in vitro* or *in vivo*) as well as the used organism and organ/ tissue/ cell line should be recorded.

In the provenance example, Lee et al. (2003) have executed *in vitro* wet-lab experiments with an egg extract of Xenopus and have shown in `WD1` that the "turnover of GSK3β, Dsh, and TCF is relatively slow" [28]. The data from this wet-lab experiment is not shown in the publication. The simulation data `SD2` contains the results of the successful validation of the simulation model `SM2`. The simulation data is presented in Figure 2 of their publication. The way the provenance graph and the metadata of `SD2` is visualized in *WebProv* can be seen in Fig 4.

**Provenance activities and relations.**   The provenance graph is formed by explicitly relating entities and activities. This is done by declaring which entities are being used or which entities are being generated by which activities. We currently distinguish four activities: building, calibrating, validating, and analyzing the simulation model.

Products of activities (i.e., entities) are connected to these activities via the relation *wasGeneratedBy*. For example, Simulation Experiments or Simulation Data may be the result of all but the Building Simulation Model activity. Activities are connected to entities via the relation *used*. For example, the Calibrating Simulation Model activity may use the Simulation Model as the object to be calibrated, some Simulation Data or Wet-lab Data for calibration, and Requirements to confirm the calibration results. All connections that we currently distinguish are shown in Table 2.

It should be noted that provenance activities in simulation studies can be defined at various levels of granularity. We have opted for a rather coarse-grained approach identifying only

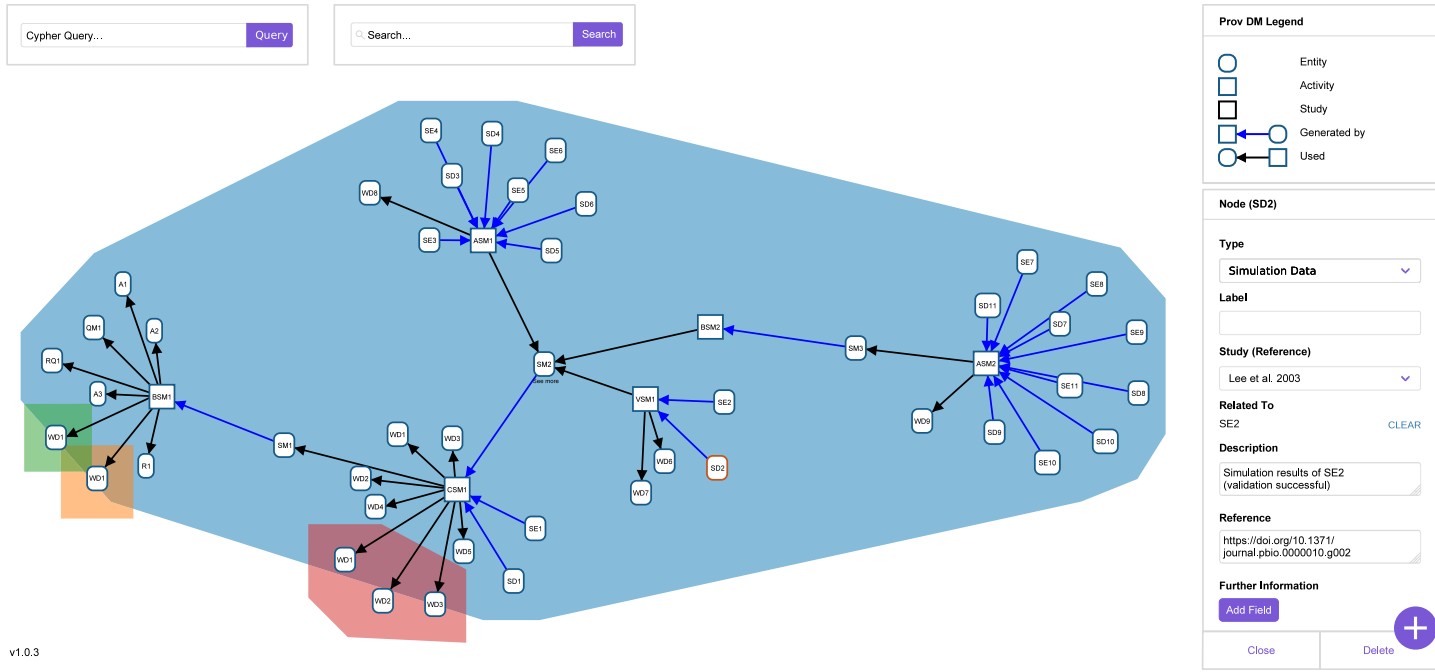

**Fig 4. Screenshot of WebProv.** This screenshot shows the provenance graph of the study by Lee et al. (2003) [28] with additional entities from three other studies [64–66], which are automatically colored differently. The node SD2 has been clicked on, which opens a box on the right with the stored and editable metadata.

crucial activities of a simulation study without explicitly denoting how an activity has used a specific entity. Thus, we aggregate activities as much as possible and leave out intermediate steps, focusing on the entities and not on the activities. From the moment provenance information is recorded automatically during the course of a simulation study, a higher level of detail could be achieved and an abstraction-based filter could be applied to zoom out to reach our granularity [2].

**Building Simulation Model** [BSM]:

The Building Simulation Model activity, also called model derivation [86], can use all entity types as any entity described above can contribute to the model building process, but it needs to have at least one link to a Research Question or Simulation Model. The only result of the building simulation activity is a Simulation Model entity. Not every update of a simulation model within a simulation study will be reflected in the provenance graph—only those changes to the model that are considered essential by the authors.

In our provenance graph of the study of Lee et al. (2003), two Building Simulation Model activities are shown. BSM1 is using wet-lab data, the research question, the qualitative model, a requirement and assumptions to develop a "provisional reference state model" [28], which forms the not yet calibrated simulation model in the study. The Building Simulation Model activity BSM2 extends the simulation model SM2.

**Calibrating Simulation Model** [CSM]:

The calibration of a simulation model is used to determine parameter values. Sometimes, switching parts of a model on or off (e.g., individual rules or model components) or choosing between entire models can also be interpreted as a discrete parameter value to be determined using methods of model selection [87]. This activity uses a Simulation Model and typically needs reference data (Wet-lab Data or Simulation Data) for the parameter estimation procedure and produces a specification or documentation of a Simulation Experiment as well as a

Simulation Data entity. If the calibration is successful, the result of this activity will always be a (calibrated) Simulation Model. Ideally, it also takes an explicit requirement into account, which, in some cases, if formally defined, can also be used for calibrating the simulation model [88, 89]. It may also use an Assumption.

In the case of the activity `CSM1` from Lee et al. (2003), several wet-lab data from their own experiments (`WD1–WD5`) as well as from Salic et al. (2000) [66] (`WD1–WD3`) are used during the calibration of the model `SM1` which produces a Simulation Experiment `SE1`, the corresponding Simulation Data `SD1` as well as the calibrated Simulation Model `SM2`.

**Validating Simulation Model** [`VSM`]:

The validation of a simulation model is used to test its validity (with regard to some requirements). Unlike calibration activities, here, the result is typically a binary answer, yes or no, which may be determined based on a specific distance measure and error threshold. The activity uses a Simulation Model and traditionally relies on reference data (Wet-lab Data or Simulation Data) that has not been used for calibration. For example, the Simulation Data from other studies may be used for the intercomparison of simulation models when performing an equal simulation experiment. Additionally, the required behavior can be formally specified in a Requirement (e.g., in a temporal logic) and automatically be checked [90, 91]. The Validating Simulation Model activity may also use an Assumption. It produces at least one entity of type Simulation Experiment as well as the corresponding Simulation Data entity. The Simulation Data entity of validation experiments stores the information whether the validation has been successful or not.

In our provenance example of Lee et al. (2003), the simulation model `SM2` is validated in the activity `VSM1` by comparing it with their own *in vitro* measurements shown in `WD6` and `WD7`. Neither a distance measure nor an explicit requirement are mentioned.

**Analyzing Simulation Model** [`ASM`]:

Similar to validation and calibration experiments, this activity provides insights into the simulation model and thus also into the system under study. The activity uses a Simulation Model and creates a Simulation Experiment as well as resulting Simulation Data. The use of Assumptions, Simulation Data, or Wet-lab Data is optional and might give an indication about the purpose of an analysis. The Analyzing Simulation Model activity aggregates all simulation experiments that are not explicitly aimed at calibration and validation.

Lee et al. (2003) analyze both the calibrated simulation model `SM2` as well as the extended simulation model `SM3` by applying parameter scans, perturbations, and sensitivity analyses which results in the Simulation Experiment `SE3–SE11` and Simulation Data `SD3–SD11`.

**Extensibility and applicability of the approach.** All of these entities, activities and relations show major steps of the development of a simulation model and, as we will see in the following section, help to interrelate different simulation studies. Still, PROV-DM would allow even more details. We have not yet considered the type `agent` from PROV-DM in our approach because this information appeared less relevant in the analyzed simulation studies. In the future, the provenance information could include the name of the agent an entity is attributed to, the agent an activity is associated with, or the name of the agent another agent acted on behalf of [19]. This would be of particular relevance if models are not only validated but also accredited, which typically involves a different group of people other than those who have developed the simulation model [92].

We have also not included the direct connection between two activities or two entities, such as the possibility to have a model being derived from another model. Thus, we have not included the following relations: a) `WasInformedBy`, which relates an activity to another one and b) `WasDerivedFrom`, which describes a direct transformation (update) of an entity into a new one. However, these relations can partly be inferred via the existing relations. For

example, a simulation model that has been generated by a Building Simulation Model activity that used another simulation model indicates that the former has been derived from the latter. Additionally, a validation activity that failed and that is followed by a Building Simulation Model activity obviously holds some information for the latter.

In our experience, it is best to capture provenance information manually or (semi-)automatically during the modeling (and simulation) process. This could be done, for example, within an artifact-based workflow system [2]. However, this would rely on a fixed life cycle definition (i.e., constraints regarding the allowed activities). Other approaches are based on automatically analyzing scripts, either with [93] or without [94] the help of user annotations. The latter has potential for a fully automatic and transparent provenance capture, however, it is difficult to implement, highly system-dependent, and only accounts for provenance information contained explicitly in the scripts, thus leaving out important information, such as research question, assumptions, or qualitative models. *WebProv*, on the contrary, is a standalone tool that works system-independently. Although it requires user input or a valid JSON input file, it also provides great flexibility regarding the information to be captured.

## Exploring the provenance of and among the Wnt simulation models of 19 simulation studies

Based on the entities and activities that were identified and defined above, we have recorded the provenance information from the 19 studies shown in bold in Table 1 as well as the provenance of entities from other publications that were used by the 19 studies. The references to the additionally used studies are found in S1 Appendix. The complete provenance information is presented in S1 Data. Screenshots and presentations of the provenance information can also be found on GitHub. We will now discuss the observations we have made during the process of capturing the provenance information and later show how the studies and simulation models relate to each other.

**Provenance of individual Wnt simulation models.** It is important to remember that we have manually collected all provenance information (entities, activities and relations), as described in the Materials and methods section. Collecting this information based on publications only is a demanding task and requires some interpretation, as natural language descriptions tend to be ambiguous. Also, the nonlinearity of the text—it is not a lab protocol after all—makes it hard to identify the relations between the entities and activities as well as the order of their execution. This would likely hamper an effective use of text mining or machine learning methods to complement or replace the manual work. Therefore, provenance information should be collected during the simulation study and ideally without an intervention of the modeler.

The Research Question was usually repeated within the abstract and throughout the introduction and discussion sections. Sometimes, there was more than one research question to be answered. In this case, we have combined these into a single entity.

Many Assumptions were introduced by the word "assume" or its derivatives. Other expressions such as "hypothesis", "is believed", "consider", "approximate", "simplify", "suggest", "suppose", or "propose" were also used by the authors to mark an assumption. However, not every sentence containing one of these words was identified to be an assumption of the simulation model. Occasionally, there were also assumptions which did not use one of the key words from above. Furthermore, two out of 19 studies did not explicitly state assumptions ([29, 53]). Generally, identifying assumptions involves many uncertainties. On the one hand, the authors might not have stated all assumptions made during the derivation of the simulation model. On the other hand, we could have easily missed an assumption or interpreted statements as

assumptions that were not meant as such by the authors. Consequently, the assumptions might look different if the authors had defined them themselves.

In order to further analyze the assumptions, we categorized them using the Systems Biology Ontology (SBO) [68]. However, the assumptions could not always be unambiguously matched to an SBO vocabulary and some assumptions dealing with biological mechanisms are not covered by SBO. For example, an autocrine signaling assumed by Mazemondet et al. (2012) [46] cannot be expressed by SBO. Some assumptions also include more than one detail which is reflected by multiple SBO categories per assumption. In this case, the assumption entity is duplicated and every assumption entity will receive its own category. The categorization of 106 collected assumptions shows that the three most used categories of assumptions deal with kinetic constants (13 times), transport (9 times), and equivalence (8 times). The result of the categorization can be found in S1 Table.

In many cases, Requirements were not given explicitly in a formal way or even as textual descriptions. We could only identify Requirements in the publications of Lee et al. (2003) [28], Wang et al. (2013) [48], Chen et al. (2014) [49], and Haack et al. (2020) [56]. The lack of Requirements was especially obvious when calibration or validation experiments were carried out without explicitly explaining the objective function.

In the surveyed publications, it was common to include a reaction scheme of the simulation model, which we referred to in the Qualitative Model entities. When recording all species, we disregarded di- or multimeric compounds established by monomers already mentioned. We also ignored different states of the species (e.g., phosphorylation states). In all provenance graphs but the one by Mirams et al. (2010) [41], at least one Qualitative Model was used by a Building Simulation Model activity to produce the first Simulation Model. Mirams et al. (2010) have directly worked with the simulation model SM2 from Lee et al. (2003).

The Simulation Models were either part of the manuscript or, more often, part of the supporting material. In 13 out of 19 cases, it was a system of ordinary differential equations. There were two simulation studies using PDEs and four using a rule-based approach (see Table 1). Although the Wnt signaling pathway has also been used to illustrate features of rule-based modeling [95, 96], only few simulation models have been developed based on a rule-based approach. The reason for this might be partly because support for thorough experimentation with rule-based models including calibration and validation has only become available during the last decade [97–99].

We categorized all 145 Simulation Experiments that we found depending on which experiment type they served (see UML class diagram shown in Fig 2). The results of the categorization are shown in Fig 5 and the details in S2 Table. Most Simulation Experiments were parameter scans, followed by time course analyses and perturbations. None of the 19 simulation studies have used steady-state analysis alone without applying another type of experiment at the same time, which we then recorded because it was more specific. The detection of steady states is typically part of an optimization, parameter scan, perturbation, and sensitivity analysis, because steady-state values are often the starting and end point of each simulation and are used for calculations.

Sometimes, simulation or wet-lab experiments have been conducted, however, the corresponding Simulation Data or Wet-lab Data is not shown. Instead, they are briefly described, and thus are without references in the provenance graphs. Authors often refer to this by stating "data not shown". For example, Lee et al. (2003) state that unpublished measurements showed that the "turnover of GSK3β, Dsh, and TCF is relatively slow" [28]. Usually, both Simulation Data and Wet-lab Data are shown in figures in the studies or published in tables or figures as part of the supplemental material. In recent years, more and more journals, such as PLOS Computational Biology, have been recommending (but not requiring) to adhere to checklists

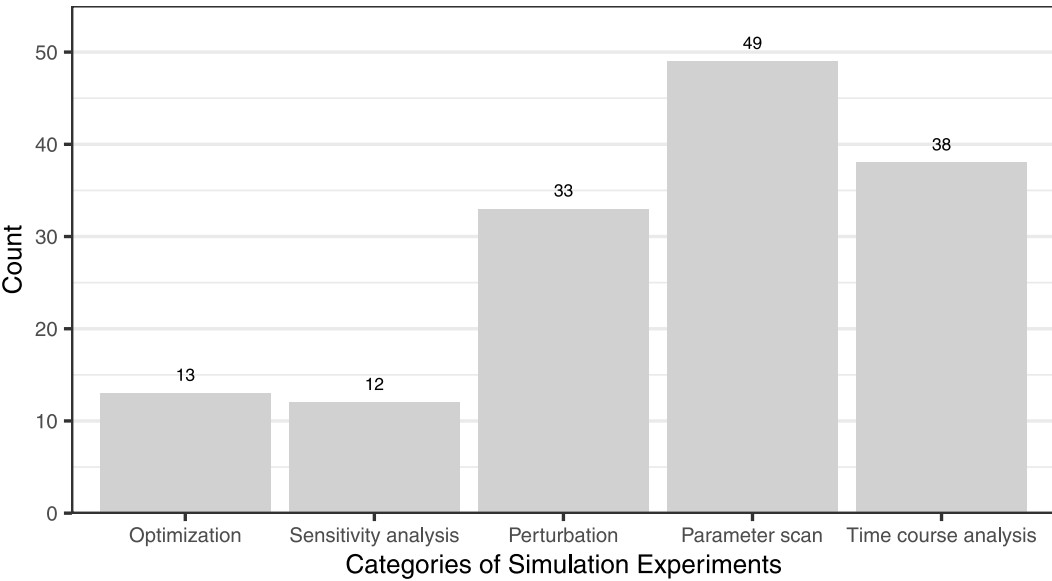

**Fig 5. Categories of the simulation experiments conducted within the analyzed 19 Wnt signaling studies.** All of these simulation experiments have been categorized. Most simulation experiments were parameter scans.

of the FAIRsharing [100] portal when reporting data, with FAIR standing for: findable, accessible, interoperable and reusable [101].

In the case of Simulation Data, the focus lies on FAIR simulation models and experiments as it should be possible to easily regenerate the data. This could be achieved, for example, by publishing a COMBINE archive [61], which is a "single file that aggregates all data files and information necessary to reproduce a simulation study in computational biology" [102]. However, the publications we have analyzed date back up to 17 years, so most data has not been published in a FAIR way.

When looking at Wet-lab Data, six out of 19 publications recorded their own wet-lab data [28, 31, 32, 45, 51, 57]. All other simulation studies either used wet-lab data from other publications or did not explicitly refer to wet-lab data at all [29, 30, 33, 34, 39, 41, 53]. The latter could only be done because the authors relied on other Simulation Models and their respective parameter and initial values. Interestingly, the wet-lab data obtained by the 19 Wnt signaling studies and the other studies which were used in the 19 studies stemmed from different organisms and cell lines. Besides human and murine cell lines (each 11 studies), xenopus, rat, hamster, and kangaroo rat were used as a model organisms. The experiments included, among others, (tumor) cell lines from the kidney (BHK, HEK293, PtK2), bone (MG-63), cervix (HeLa), brain (neural progenitor cells), and fibroblasts (L cells, NIH/3T3). All different cell lines directly used in the studies are presented by the colored rectangles in Fig 6. The cell lines and organisms that were included in the simulation studies are shown in Table 3.

When looking at the CSM and VSM activities, we detect that only a few simulation models were both calibrated and validated based on wet-lab data, namely those of Lee et al. (2003) [28], Kogan et al. (2012) [45], and Haack et al. (2015) [51]. Some authors just calibrated their simulation models ([33, 46, 48, 49, 53, 57]) and in two studies ([30, 56]) they were only validated (having assumed plausible ranges for parameter values). The simulations models developed in [29, 32, 34–36, 39, 41] had neither been calibrated nor validated explicitly, because they used parameter values from other studies. Sick et al. (2006) [31] used arbitrary parameter values in their simulation models.

**Fig 6. Provenance graph of all Wnt/β-catenin simulation studies considered here (black outlines) as well as additionally used studies (gray outlines).** The colors indicate the cell lines used in wet-lab experiments of that study (see legend below graph). Gray boxes represent pure Wnt/β-catenin simulation studies without acquiring wet-lab data. White boxes display publications used by some of the Wnt/β-catenin simulation studies that are either text books or simulation studies without published wet-lab data. The figure was created using the R package `DiagrammeR` [103]. The references to the additionally used studies are found in S1 Appendix.

Some studies develop a story line where a simulation model has been successively refined, extended, or composed ([28, 33, 34, 36, 41, 46, 48, 49, 51, 56]). These studies are characterized by a Simulation Model being used by a Building Simulation Model activity, both nodes of the same study.

Some simulation studies resulted in the development of multiple simulation models that are neither extensions nor compositions but rather form a revision or alternative to other

**Table 3. Cell lines and organisms used building the simulation models.**

| Reference | Egg extract | Embryo | Fibroblasts L cells | Fibroblasts NIH/3T3 cells | Mammary gland (C57MG) cells | Osteoblast-like cells (MG-63) | Presomitic mesoderm (PSM) cells | Skin | Cervical cancer epithelial (HeLa S3) cells | Colon carcinoma RKO cells | Embryonic kidney epithelial (HEK 293) cells | Neural progenitor (ReNcell VM) cells | Baby hamster kidney (BHK) cells | Epithelial kidney cells (PtK2) | Pheochromocytoma cells | Not applicable |
|---|---|---|---|---|---|---|---|---|---|---|---|---|---|---|---|---|
| [28] | x | | | | | | | | | | | | | | | |
| [29] | x | | | | | | | | | | | | | | | |
| [30] | x | | | | | | | | | | | | | | | |
| [31] | | | | | | | | x | | | | | | | | |
| [32] | o | | | | | | | | | | x | | | | | |
| [33] | | x | | | | | | | | | | | | | | x |
| [34] | x | | | | | | | | | | | | | | | |
| [35] | x | | | | x | | | | | | x | | | | | |
| [36] | x | | | | | | x | | | | | | | | | |
| [39] | o | | | | | | | | | | | | | | | |
| [41] | x | | | | | | | | | | | | | | | |
| [45] | x | | x | | | | | | | | | | | | | |
| [46] | x | | | | | | | | | | | x | | | | |
| [48] | o | | | | | | x | | | | | | | | | |
| [49] | x | | | | | | | | | x | x | | | | | |
| [51] | x | | x | x | | | | | | | x | x | x | x | | |
| [53] | o | | | | | | | | | | x | | | | x | |
| [56] | o | | x | x | | | | | x | | o | x | o | o | | x |
| [57] | | | | | | x | | | | | | | | | | |
| Cell line / organ | Xenopus | | Mouse | | | | | | Human | | | | Hamster | Kangaroo rat | Rat | |

The top row shows the origin of the cells used by the 19 studies. *Not applicable* means that the parameters values were obtained from theoretical calculations/considerations from a textbooks or without concrete reference to a wet-lab study. The colors denote the organisms of the cell line/organ used (see bottom row). The "x" denotes *directly used*, the "o" denotes *indirectly used by using (parts of) a referenced simulation model.*

simulation models developed in the same study: [33, 39, 41, 46, 48, 49]. This can be seen in the provenance graph of a single study when the last simulation model is not connected by a directed path to other simulation models of that study or when the simulation models are part of disjoint branches of the provenance graph. For example, in Mazemondet et al. (2012) [46], the core model of the Wnt/β-catenin signaling pathway has been calibrated with wet-lab data from Lee et al. (2003) [28], but this calibrated simulation model was not used and instead, a

new calibration with wet-lab data from another study took place. Two simulation studies ([35, 41]) show disconnected graphs. This shows that these researchers considered, built, and analyzed multiple simulation models independently of each other.

**Beyond individuals: A family of Wnt simulation models.**   Whereas before we have looked at the properties of individual simulation studies, we are now going to investigate the interrelations between the 19 Wnt/β-catenin signaling simulation models. We will identify features that transform a set of simulation models into a family.

Fig 6 shows an overview of all simulation studies considered in our research by zooming out of the individual provenance graphs. All but two simulation models are (indirectly) connected to the model proposed by Lee et al. (2003) [28], which was the first validated simulation model of the canonical Wnt signaling pathway. This shows that models are usually not built independently from one another, but are often extensions or revisions of formerly published models or use the parts of the reactions or parameter values. The two exceptions not linked to Lee et al. (2003) [28] are the simulation models by Sick et al. (2006) [31] and by Rodríguez-González et al. (2007) [33]. Sick et al. (2006) use a reaction-diffusion system (Gierer-Meinhardt equations [104]) to model the interplay between Wnt and its antagonist Dkk with just two equations. Rodríguez-González et al. (2007) consider Wnt and Axin—two key players of the Wnt/β-catenin signaling pathway—but only in the context of the Notch signaling pathway and thus the model only contains a fragment of the canonical Wnt pathway.

We have also included the cell lines/tissue used for wet-lab experiments within each study. As already seen in Table 3, we observe that different simulation studies use data or models obtained using other cell lines. This may be valid as the Wnt/β-catenin signaling pathway is evolutionarily conserved [24], which means that data can be shared. Still, care must always be taken when using, for instance, parameter values determined with one cell line in a study that uses another cell line.

When looking at the same graph using a circular layout, we observe four clusters of two or more studies, as shown in S1 Fig. We have also colored the studies according to additional pathways they include and observe that the clusters separate the studies depending on these additional cellular mechanisms. The central cluster includes the Wnt model by Lee et al. (2003) [28] as well as the studies [30, 34, 35, 41]. A second cluster forms around the simulation studies [45, 46, 51, 56, 57] and either includes the same wet-lab data from [105, 106], the Cell cycle [46] or ROS [51, 57]. A third cluster includes the pathways of Notch [33, 48] and Notch + MAPK/ERK [36]. Even though the algorithms locate [29] in the same cluster, it is content-wise rather part of the central cluster. A forth cluster forms around studies that include MAPK/ERK [32] or MAPK/ERK + PI3K/Akt [53]. All other models are not part of a cluster and are either completely disconnected from the other studies [31] or include E-cadherin and the cell cycle [39] or just E-cadherin [49].

When comparing Fig 6 and S1 Fig, we find that the wet-lab data from just three studies, namely from [105–107], have been reused by simulation studies. On the one hand, this is surprising as wet-lab data can be reused for parameter estimation or model validation. On the other hand, when authors use parts or entire simulation models published by others, they do not necessarily recite the references that were used for obtaining the parameter and initial values that come with the model. Thus, a direct connection from the new simulation model to the wet-lab data used by another simulation study is not made.

## Conclusion

Provenance of simulation models provides information about how a simulation model has been generated and about the steps and various sources that contributed to its generation.

Here, we have developed a specialization of PROV-DM focusing on entities and activities. It builds on an earlier PROV-DM specialization in which Simulation Model, Simulation Experiment, Simulation Data, and Wet-lab Data have been identified as crucial entities of simulation studies [20]. Additionally, we have taken knowledge of modeling and simulation life cycles [1] into account and identified the Research Question, Assumptions, Requirements, the Qualitative Model to be important ingredients of the provenance of simulation models. We also distinguish between Building Simulation Model, Calibrating Simulation Model, Validating Simulation Model, and Analyzing Simulation Model activities and connect the entities and activities by using the relations *wasGeneratedBy* and *used*. In our definitions of the entities and activities, we aimed at achieving the minimal level of detail, or granularity, of the provenance graph to understand the course of a simulation study. We also kept the necessary metadata of the entities and activities to a minimum to convey both the main idea of the simulation study and the content of each entity and activity. For storing, visualizing, and querying the provenance information, we have created the web-based tool *WebProv* that allows for each entity and activity to store (customized) metadata and references.

In order to examine our specialization of PROV-DM, the extensive analysis of 19 simulation studies of the canonical Wnt signaling pathway provided a suitable case study. We were able to explicitly show that most studies are connected to one or more other Wnt simulation studies, using (parts of) their simulation models, in addition to various data from wet-lab studies. Our results show the outstanding role of the Wnt simulation model by Lee et al. (2003) [28] as the origin for most other models in our survey. Thus, a family of Wnt signaling models could be revealed.

In conclusion, provenance information provides added value to the existing list of documentation requirements and could complement and enrich the effort of "harmonizing semantic annotations for computational models in biology" [108]. Together with the exploitation of community standards and ontologies, provenance information opens up further possibilities of reusing and analyzing simulation models, for example, to help with model selection, model merging, or model difference detection. Of course, to be fully accepted, our specialization of PROV-DM should be subject to a standardization initiative. We think that *WebProv*, or a similar tool, would be a valuable extension to model repositories such as BioModels, as one could see where a simulation model comes from, whether there are other models connected to it, and in which way they are connected. This would help to quickly interpret the increasing number of published simulation models and find a suitable one for your research.

## Supporting information

**S1 Appendix. Additional references for entities used by Wnt simulation studies.** We show the references to additional studies that contain entities used by some of the 19 Wnt simulation studies.
(PDF)

**S1 Data. Complete provenance information from 19 Wnt simulation studies.** This file contains the provenance information from the 19 analyzed simulation studies of the Wnt signaling pathway. It was exported from *WebProv* and may be imported into another instance of the tool.
(JSON)

**S1 Fig. Provenance graph of all 19 Wnt/β-catenin simulation studies and their depending studies using a circular layout.** Studies which include additional pathways have been colored.
(PDF)

**S1 Table. Categorized assumptions.** We present the results of the categorization of all assumptions found in the 19 simulation studies using SBO. We have have also added information about the key words that accompanied the assumptions.
(CSV)

**S2 Table. Categorized simulation experiments.** We present the results of the categorization of the simulation experiments found in the 19 simulation studies using our categories.
(CSV)

## Acknowledgments

We thank Nadja Schlungbaum for her excellent technical support.

## Author Contributions

**Conceptualization:** Kai Budde, Pia Wilsdorf, Adelinde M. Uhrmacher.

**Data curation:** Kai Budde.

**Funding acquisition:** Adelinde M. Uhrmacher.

**Investigation:** Kai Budde.

**Methodology:** Kai Budde, Pia Wilsdorf, Fiete Haack, Adelinde M. Uhrmacher.

**Project administration:** Kai Budde, Adelinde M. Uhrmacher.

**Software:** Jacob Smith.

**Supervision:** Adelinde M. Uhrmacher.

**Validation:** Fiete Haack.

**Visualization:** Kai Budde, Fiete Haack.

**Writing – original draft:** Kai Budde, Pia Wilsdorf, Adelinde M. Uhrmacher.

**Writing – review & editing:** Kai Budde, Jacob Smith, Pia Wilsdorf, Fiete Haack, Adelinde M. Uhrmacher.

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
