## [Decision Letter · Decision Letter 0]

30 Mar 2021

Dear Mr. Budde,

Thank you very much for submitting your manuscript "Relating simulation studies by provenance—Developing a family of Wnt signaling models" for consideration at PLOS Computational Biology.

As with all papers reviewed by the journal, your manuscript was reviewed by members of the editorial board and by several independent reviewers. In light of the reviews (below this email), we would like to invite the resubmission of a significantly-revised version that takes into account the reviewers' comments.

We cannot make any decision about publication until we have seen the revised manuscript and your response to the reviewers' comments. Your revised manuscript is also likely to be sent to reviewers for further evaluation.

Sincerely,

Pedro Mendes, PhD

Associate Editor

PLOS Computational Biology

Jason Haugh

Deputy Editor

PLOS Computational Biology

Reviewer's Responses to Questions

**Comments to the Authors:**

Reviewer #1: Much has been made recently of the importance of reproducibility of scientific research. In biological simulation studies, such as this paper considers, this means providing the models and the analysis instructions in standard machine readable forms. This paper takes this a step further to look at the issue of provenance for these models. Namely, how are models related to previous models and experimental studies. In particular, the authors looks at a family of 19 models of the Wnt signaling pathway, which that manually link together using the PROV-DM ontology. To construct these relationships, they have developed a web tool, WebProv, to link studies with PROV-DM types and relations. As the authors point out, extracting these relationships from published studies is a highly laborious process that requires many assumptions along the way. Ideally, in the future, these provenance networks should be developed at the time of model construction during the simulation study.

This paper is an important demonstration of both the process of creating and utility of provenance networks for simulation studies. The prototype software tool presented should help facilitate future such activities. This proof-of-concept presented in this paper should become a tutorial for others that would undertaken this task for their models and simulation studies. The key issue remaining is how to motivate and facilitate others to create this information. This is not a problem that the authors can solve, but rather one that the community and the journal publishers should devote time and energy to in order to further improve the reproducibility of science.

Reviewer #2: This is an interesting paper that looks in detail at 19 Wnt related modeling papers. As a practicing modeler myself the most interesting pages started on page 14 (Provenance of individual Wnt simulation models) which discusses the various issues encountered, problems with the current ontolgies etc. I think this is the most important part of the paper from the point of view of the plos comp bio readership, such analyses have not been done as exhaustively as this one.

Some of the material, particular the descriptions of the provenance entries (which seems to dominate the paper) could be summarized in a table and the textual component moved to an appendix. This would allow the reader to get straight to the most interesting papers of the paper.

Caption: The captions to some of the figures could be improved.

Fig 2: This caption starts with the word ‘additionally’ which doesn’t sit well. Also the caption is too short, given that this is probably one of the more important figures. It took me a while to realize what the terms ASM, CSM etc meant (They were in Table 2). I would spell out these abbreviations (ASM, CSM, VSM, BSM) in the caption (Table 2 can remain unchanged), this will save the reader for having to search for their meaning. The caption would also add one sentence on how to read the figure. I know that earlier on the authors explain what an arrow means but

that was some pages away and since plos comp bio are generally no computer scientists I would add that explanation of the arrow to the caption as well.

Fig 3: In general, notation used in UML diagrams is not familiar to most modelers but in this case the diagram looks simple enough that its seems fairly self-explanatory. No action required.

Minor: Typo in caption first sentence : ‘prociding’, I’m not sure what that word means, probably a typo but not sure what word should be there instead?

Software: As it stands it is likely that very few people will use WebProv, the reason is that it requires far too much work to install, plus does it also require a backend sever?

The tool looks useful so why make it difficult to get hold of?

What I would recommend is move everything if possible to the client (including the database which doesn’t seem large) and host it as a github web page project so that when a user clicks on the url the application will show up, no installation necessary (which I think is one of the main attractions of web software) – see https://pages.github.com/. I strongly recommend something like this otherwise your work will not have the impact it should.

Documentation: There is no easily accessible documentation for the software. It looks like users are expected to download the github repo then select the indexl.html file in Docs. It would be better to host a proper (eg readthedocs) documentation on the github account itself. The only documentation link in the readme take a user to process manger 2 page.

Minor:

1. Introduction, second paragraph, first sentence, ‘conduction’, is that the right word?

2. There is no reference to the youtube video on page 4 (footnote 2), also put the github ural there as well since the github repository is mentioned.

Reviewer #3: In this manuscript, the authors present a continuation of previous work to make use of the PROV-DM to represent provenance of biosimulation studies. The authors then use this data model to encode the provenance of a number of Wnt signalling models from the literature, and use the encoded provenance knowledge to examine the relationships between these published studies. The authors have also developed an open-source web-based tool providing a graphical user interface for creating, editing, exploring, and searching the provenance knowledge encoded in this manner. This seems to be a very useful approach for capturing provenance knowledge of systems biology modelling studies and appears to be extensible to capture richer provenance semantics as the collection/recording methods improve in future and this approach is applied in different domains or different types of models.

The authors should be commended for making all the software and data used in this manuscript freely available and documented in a manner sufficient to enable others to repeat the analysis presented here.

I suggest the entire manuscript is thoroughly proof-read as some of the grammar and word choices are a bit unusual. "cell biological systems" in the abstract is one example that could be tidied up.

As the authors state, the knowledge they have extracted from the literature and encoded in the example provenance graph used in this work makes a useful contribution to the community of potential users of these Wnt signally models. I wonder if the authors have any plans or thoughts on the integration of this knowledge into a community repository, perhaps in a way that others could contribute to? For the subset of models that are available in the Biomodels database, for example, could the provenance knowledge be contributed back to the database?

Following that thought, some of the provenance knowledge captured here is similar to that represented in the Biomodels database using the "isDerivedFrom" predicate in the SBML model annotations (see for example the analysis of diabetes models presented in https://dx.doi.org/10.1038%2Fpsp.2013.30). Have the authors compared this knowledge for the subset of Wnt models available in the Biomodels database to see if similar (although less semantically rich) patterns of model evolution are present to their analysis presented in this manuscript?

Using the SBO to annotate the assumptions seems an odd choice to me. Looking at Table S1, it seems that the SBO terms are giving a very high level annotation as to the type of model entity mentioned in the assumption, but doesn't provide any semantics about what the assumption is. Looking at assumptions annotated with SBO:0000009 (kinetic constant), for example, a user can search for assumptions that have something to do with a kinetic constant, but doesn't help to examine if its an assumption based on time scale analysis (e.g., row 3) or perhaps just an assumption that certain behaviour is assumed (e.g., row 13). I wonder if something like the Evidence and Conclusion Ontology (https://evidenceontology.org/) might provide a source of more meaningful terms to use in annotating assumptions? I may simply be missing something here, so perhaps a bit more explanation about how the SBO annotations are being used to annotate assumptions would help clarify things (or future work to extend the current work with enriched semantics?).

The authors define a minimal set of PROV-DM entities and activity types they have found useful for capturing provenance information of simulation studies when extracting provenance knowledge from the published literature. This minimal set does seem sufficient for the Wnt signalling demonstration presented here and the authors briefly explore how this set could be expanded in future. But I worry that the wet-lab data entity seems under-specified and perhaps less useful than it could be. While I understand that often in the literature the source of experimental data is not clearly described, with the recent growth of platforms like https://www.protocols.io/ which enable scientists to provide rich descriptions of their protocols in a reusable manner, I wonder if the authors have considered how to incorporate that type of knowledge into their provenance graphs?

Minor comments

--------------

It may not be obvious to the reader exactly what PROV is when first mentioned in the abstract.

The assertion in the abstract that this provenance information is all that is required to answer the question of an "appropriate starting point" is perhaps overstating things. The provenance information contributes to that answer, but it is not the only knowledge that is required to make an informed decision.

Figure 3 caption: "prociding" - perhaps meant to be providing?

I completely agree with the authors that provenance information should be collected during the simulation study, but I wonder if the authors have given any thought to how their WebProv tool could be utilised as part of a typical modelling lifecycle to help encourage modellers to do so?

**Have all data underlying the figures and results presented in the manuscript been provided?**

Reviewer #1: Yes

Reviewer #2: Yes

Reviewer #3: Yes

PLOS authors have the option to publish the peer review history of their article (what does this mean?). If published, this will include your full peer review and any attached files.

Reviewer #1: **Yes: **Chris J. Myers

Reviewer #2: No

Reviewer #3: **Yes: **David P Nickerson
---

## [Decision Letter · Decision Letter 1]

29 Jun 2021

Dear Mr. Budde,

We are pleased to inform you that your manuscript 'Relating simulation studies by provenance—Developing a family of Wnt signaling models' has been provisionally accepted for publication in PLOS Computational Biology.

Best regards,

Pedro Mendes, PhD

Associate Editor

PLOS Computational Biology

Jason Haugh

Deputy Editor

PLOS Computational Biology

Reviewer's Responses to Questions

**Comments to the Authors:**

Reviewer #2: None

Reviewer #3: I thank the authors for their detailed response to my orginal review.

**Have the authors made all data and (if applicable) computational code underlying the findings in their manuscript fully available?**

Reviewer #2: Yes

Reviewer #3: None

PLOS authors have the option to publish the peer review history of their article (what does this mean?). If published, this will include your full peer review and any attached files.

Reviewer #2: No

Reviewer #3: **Yes: **David P. Nickerson

---

## [Editor Report · Acceptance letter]

23 Jul 2021

PCOMPBIOL-D-21-00298R1 

Relating simulation studies by provenance—Developing a family of Wnt signaling models

Dear Dr Budde,

I am pleased to inform you that your manuscript has been formally accepted for publication in PLOS Computational Biology. Your manuscript is now with our production department and you will be notified of the publication date in due course.

With kind regards,

Andrea Szabo
